# Vortex states in an acoustic Weyl crystal with a topological lattice defect

Qiang Wang [1], Yong Ge[2], Hong-xiang Sun [2], Haoran Xue [1], Ding Jia[2], Yi-jun Guan[2], Shou-qi Yuan [2✉], Baile Zhang [1,3✉] & Y. D. Chong [1,3✉]

Crystalline materials can host topological lattice defects that are robust against local deformations, and such defects can interact in interesting ways with the topological features of the underlying band structure. We design and implement a three dimensional acoustic Weyl metamaterial hosting robust modes bound to a one-dimensional topological lattice defect. The modes are related to topological features of the bulk bands, and carry nonzero orbital angular momentum locked to the direction of propagation. They span a range of axial wavenumbers defined by the projections of two bulk Weyl points to a one-dimensional subspace, in a manner analogous to the formation of Fermi arc surface states. We use acoustic experiments to probe their dispersion relation, orbital angular momentum locked waveguiding, and ability to emit acoustic vortices into free space. These results point to new possibilities for creating and exploiting topological modes in three-dimensional structures through the interplay between band topology in momentum space and topological lattice defects in real space.

[1] Division of Physics and Applied Physics, School of Physical and Mathematical Sciences, Nanyang Technological University, Singapore, Singapore. [2] Research Center of Fluid Machinery Engineering and Technology, School of Physics and Electronic Engineering, Jiangsu University, Zhenjiang, China. [3] Centre for Disruptive Photonic Technologies, Nanyang Technological University, Singapore, Singapore. ✉email: shouqiy@ujs.edu.cn; blzhang@ntu.edu.sg; yidong@ntu.edu.sg

Topological lattice defects (TLDs) are crystallinity-breaking defects in lattices that cannot be eliminated by local changes to the lattice morphology, due to their nontrivial real-space topology[1]. Although they give rise to numerous important physical effects in their own right[2], TLDs can have especially interesting consequences in materials with topologically nontrivial bandstructures[3–8]. For instance, Ran et al. have shown theoretically that introducing a screw dislocation into a three dimensional (3D) topological band insulator induces the formation of one-dimensional (1D) helical defect modes, which are protected by the interplay between the Burgers vector of the defect and the topology of the bulk bandstructure[5]. Aside from topological band insulators[5,9–12], other topological phases are predicted to have their own unique interactions with TLDs, including Weyl semimetals, topological crystalline insulators, and higher-order topological insulators[13–18]. TLD-induced modes provide a way to probe bandstructure topology independent of standard bulk-boundary correspondences[6–8,11,15,17], and may give rise to exotic material properties such as anomalous torsional effects[13]. Experimental confirmations have, however, been hampered by the difficulty of accessing TLDs in real topological materials[19–21].

Recently, various groups have turned to classical wave metamaterials[22–26] to perform the experimental studies of the interplay between TLDs and topological bandstructures, including the demonstration of topologically-aided trapping of light on a dislocation[23], robust valley Hall-like waveguiding along disclination lines[24], and defect-induced fractional modes[25,26]. The preceding studies have all been based on two dimensional (2D) lattices; 3D lattices with TLD-induced topological modes have thus far only been investigated theoretically.

Here, we design and experimentally demonstrate a 3D acoustic metamaterial that hosts topological modes induced by the presence of a TLD. Without the TLD, the bulk metamaterial forms a Weyl crystal, whose 3D bandstructure contains topologically nontrivial degeneracies called Weyl points[27–41]. Weyl crystals are known to exhibit, along their 2D external surfaces, Fermi arc states that are protected by the topology of the Weyl points[32,33]. The introduction of the defect generates a family of modes localised to the line of the TLD (in real space). Moreover, in a manner analogous to the formation of regular Fermi arcs, the modes span the projections of two Weyl points of opposite topological charge in the axial momentum space $k_z$. The TLD-bound modes for each $k_z$ can be interpreted as a 2D bound state generated by a strongly localised pseudo-magnetic flux associated with the TLD, in accordance with earlier theoretical predictions about disclinations in 2D topological materials[8]. Hence, these modes arise from the interplay between the TLD and the 3D Weyl bandstructure. The TLD-bound modes carry nonzero orbital angular momentum (OAM), locked to their propagation direction. For each $k_z$, the sign of the OAM depends on the Chern number of the 2D projected band structure, and matches the chirality of the robust localised state that appears in a Chern insulator on a 2D surface with singular curvature[42–45]—a prediction that has never previously been verified in an experiment[8,46,47]. To our knowledge, this is also the first demonstration of a 3D topology-induced mode carrying nonzero OAM. Classical waves with nonzero OAM have a variety of emerging applications including vortex traps and rotors[48,49] and OAM-encoded communications[50]. Although chiral structures have previously been studied for the purposes of OAM waveguiding, those waveguides support multiple OAM modes with different propagation constants[51]; by contrast, the present topological waveguide supports, for each $k_z$, a single robust bound mode with nonzero OAM.

## Results

**Design of the Weyl acoustic structure.** The emergence of a TLD-bound topological mode is conceptually illustrated in Fig. 1a. In a Weyl semimetal, topologically-charged Weyl points in the 3D bulk imply the existence of Fermi arc modes on 2D external surfaces of the crystal. In the 2D surface momentum space, each Fermi arc extends between the projections of two oppositely-charged Weyl points. The introduction of a TLD into the Weyl crystal breaks translational symmetry in the $x$–$y$ plane while maintaining it along $z$, and generates modes that are spatially localised to the 1D string formed by the TLD. Viewed from momentum space, the TLD-bound modes extend between the projections of the two Weyl points into the 1D momentum space $k_z$.

Recently, the discovery of higher-order topological materials[52] has led to the idea of higher-order Weyl and Weyl-like phases[53–61], which can host "higher-order Fermi arcs"[56–58,61]. Like the TLD-bound modes discussed in this paper, higher-order Fermi arc modes are one-dimensional, but they arise from a completely different mechanism involving higher-order topological indices[56,57,61]. Moreover, they lie along external hinges, whereas the present TLD-bound modes are localised to the line of the TLD, embedded inside a 3D bulk.

We designed and fabricated a 3D acoustic crystal formed by chirally structured layers stacked along $z$, as shown in Fig. 1b–e. Without any TLD, an $x$–$y$ cross section of the structure would form a triangular lattice. The TLD is introduced by a "cut-and-glue" procedure in which a $\pi/3$ wedge is deleted (Fig. 1c inset) and the edges are reattached by deforming the rest of the lattice (see Methods). The experimental sample is formed by stacking 3D-printed structures, with a total of 21 layers (see Methods); a photograph is shown in Fig. 1e.

The 3D Brillouin zone of the acoustic crystal, in the absence of the TLD, is depicted in the left panel of Fig. 2a. Weyl points exist at $K$ and $K'$ ($H$ and $H'$), with topological charge $+1$ $(-1)$[39,62,63]; for details, refer to Supplementary Note 1. Consider the Weyl point at $K$ or $K'$ (the analysis for $H$ and $H'$ is similar). In its vicinity, the wavefunctions are governed by the effective Hamiltonian

$$\mathcal{H} = -i(\tau_z \sigma_x \partial_x + \sigma_y \partial_y) + k_z \tau_z \sigma_z, \tag{1}$$

where $\tau_i$ ($\sigma_i$) denotes valley (sublattice) Pauli matrices, we have rescaled each spatial coordinate so that the group velocity is unity, and $k_z$ is the wavenumber in the $z$ direction.

**Pseudo-magnetic flux of the lattice defect.** With the introduction of the TLD, $k_z$ remains a good quantum number; in the $x$–$y$ plane, the distortion introduced by the TLD can be modelled as a matrix-valued gauge field[8] that mixes the valleys (i.e., $K$ with $K'$ and $H$ with $H'$). The effective Hamiltonian can be brought back into block-diagonal form by a unitary transformation[8], whereby the Hamiltonian for each block has the form of Eq. (1) but modified by

$$\tau_z \to \tau', \quad \nabla \to \nabla + i\tau'\mathbf{A}, \quad \mathbf{A} = (4\Omega r)^{-1}\mathbf{e}_\theta, \tag{2}$$

where $\tau' = \pm 1$ is the block index, $r$ is the radial coordinate and $\mathbf{e}_\theta$ is the azimuthal unit vector in the unfolded space, and the factor $\Omega = 5/6$ is the number of undeleted wedges. Unlike previously studied strain-induced pseudo-magnetic fluxes in Weyl semimetals[63–65], the pseudo-magnetic flux here is strongly localised[8,66]. Moreover, unlike previous studies of pseudo-magnetic fluxes generated by screw dislocations, the pseudo-magnetic flux is $k_z$-independent[5,13].

Viewed from 2D, the pseudo-magnetic flux induces topologically protected chiral defect states. For each $k_z > 0$, one can show[5,8] that there is a single bound solution (among the two Weyl Hamiltonians) localised at $r = 0$. This remains true even

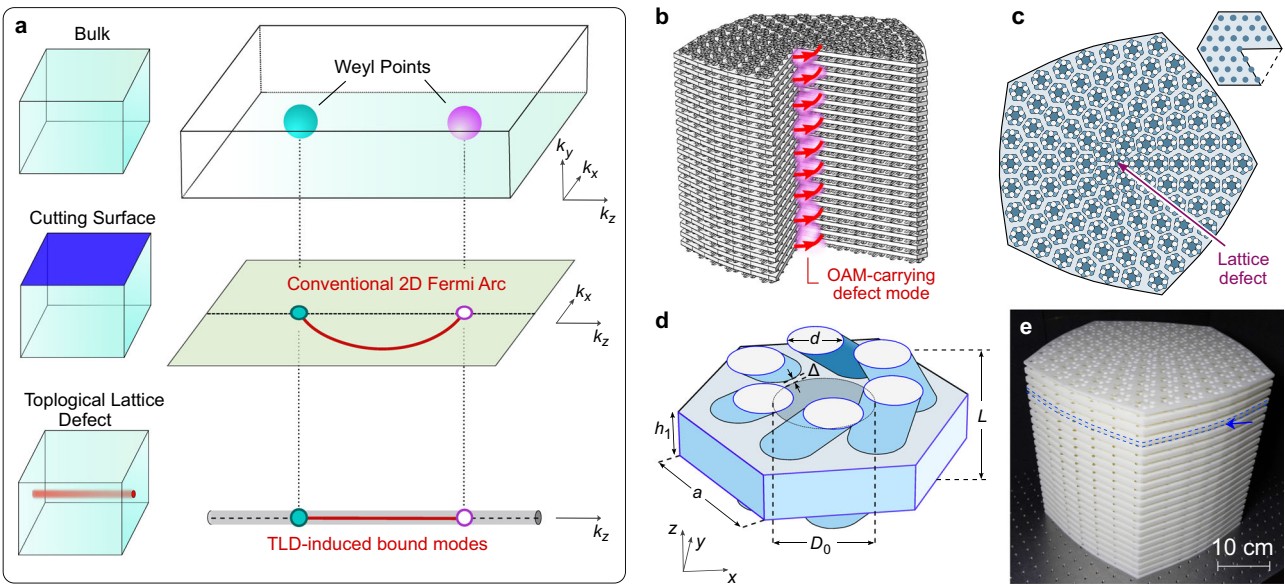

**Fig. 1 Weyl acoustic structure with a topological lattice defect (TLD). a** Conceptual illustration of the formation of a mode bound to the TLD. Top row: a bulk crystal (left) hosts two Weyl points with opposite topological charges in 3D momentum space (right). Middle row: truncating the crystal (left) creates a Fermi arc extending between the projections of the Weyl points in the 2D surface momentum space (right). Bottom row: adding a TLD (left) results in bound modes that extend between the projections of the Weyl points into the 1D momentum space (right). **b** Schematic of an acoustic lattice made of chiral layers with a central TLD, stacked along $z$. A section is omitted to show the internal structure. TLD-bound mode carrying nonzero orbital angular momentum (OAM) propagates along the TLD (pink region). **c** Top-down schematic of one layer. The TLD is generated by deleting a $\pi/3$ wedge from a triangular lattice (inset). **d** Close-up view of one unit cell. The periodicity in $z$ is $L = 1.8$ cm. Each unit cell consists of an air-filled sheet of thickness $h_1 = 0.8$ cm, with a central solid rod of diameter $D_0 = 1.6$ cm ringed by six skewed air-filled tubes of diameter $d = 0.9$ cm in a hexagonal arrangement. The tubes advance by $\pi/6$ around the hexagon between layers. The spacing between the central rod and the tubes is $\Delta = 0.1$ cm, and the side length of the hexagonal cell is $a = 4/\sqrt{3}$ cm. All other regions are solid resin. **e** Photograph of the experimental sample with 21 layers. Blue dashes indicate one of the layers, and the arrow indicates one of the gaps for inserting acoustic probes.

when $k_z$ is non-perturbative. For fixed $k_z$, the lattice in the absence of the TLD maps to a 2D Chern insulator whose Chern numbers switch sign with $k_z$ (the gap closes at 0 and $\pm\pi/L$); upon introducing the TLD via the cut-and-glue construction, one of the two sub-blocks in the effective Hamiltonian ($\tau' = 1$ for $0 < k_z < \pi/L$, and $\tau' = -1$ for $-\pi/L < k_z < 0$) exhibits a solution that is localised to the TLD[8]. As we vary $k_z$, this family of solutions spans the projections of the Weyl points at $K(K')$ and $H(H')$. Note that the overall acoustic structure preserves time-reversal symmetry ($T$), but the individual Hamiltonian sub-blocks effectively break $T$; the defect mode at $-k_z$ thus serves as the time-reversed counterpart of the defect mode at $k_z$, with opposite chirality. For further details, refer to Supplementary Note 2.

The upper panel of Fig. 2b shows the numerically computed acoustic band diagram for the TLD-free bulk structure, projected onto $k_z$. The relevant bands along $K$-$H$ ($M$-$L$) are plotted in green (orange), and the gap region is shown in white. The lower panel of Fig. 2b shows the corresponding band diagram for a structure with a TLD, which is periodic along $z$ and has the same $x$–$y$ profile as the experimental sample (Fig. 1b–e). These numerical results reveal the existence of TLD-bound modes, plotted in red, which occupy the gap and span almost the entire $k_z$ range. (Near $k_z = 0$ and $k_z = \pi/L$, they are difficult to distinguish from bulk modes due to finite-size effects.)

In Fig. 2c,d, we show the mode distributions for the TLD-bound modes at $k_z = \pm 0.5\pi/L$. The modes are strongly localised to the center of the TLD; their intensity profiles are identical since the two modes map to each other under time reversal. The phase distributions (inset) reveal that the $k_z > 0$ ($k_z < 0$) TLD-bound mode has winding number $+1$ ($-1$). This winding number is tied to the Chern number of the 2D projected band structure for fixed $k_z$. The fact that the TLD-bound modes carry nonzero OAM,

locked to the propagation direction, distinguishes them from previously studied topological defect modes[67–70] and hinge modes[56,57,61] that have zero OAM. Moreover, we have verified numerically that the TLD-bound modes' localisation and OAM are robust to in-plane disorder, consistent with their topological origin (see Supplementary Note 3).

**Spectrum and field distribution measurements**. We performed a variety of experiments to characterise the TLD-bound modes in the fabricated structure. First, we investigated their dispersion curve by threading an acoustic source into the bottom layer of the sample, near the center of the TLD. A probe is inserted into the other 20 layers in turn, via the central air sheet in each layer, as indicated by the blue arrow in Fig. 1d. The acoustic pressure, measured close to the center of the TLD, is Fourier transformed to obtain the spectral plot shown in Fig. 3a. The overlaid red dashes are the numerically obtained TLD-bound mode dispersion curve (Fig. 2b), which closely matches the intensity peaks in the experimental results. We then repositioned the source and probe away from the TLD, obtaining in the spectrum shown in Fig. 3b; this matches the bulk spectrum obtained numerically, with the spectral intensities peaking in the bulk bands. For details about the source and probe positions, see Supplementary Note 4.

The acoustic pressure intensity at $k_z = \pi/2L$ is plotted versus frequency in Fig. 3c. A narrow peak corresponding to the TLD-bound modes is clearly observable within the bulk gap, with only a small frequency shift of 80 Hz relative to the numerically predicted eigenfrequency. For excitation near the TLD, the measured intensity distribution at frequency $f = 4.924$ kHz is plotted in Fig. 3d, showing strong localisation around the TLD. The radial dependence of the intensity distribution is plotted in Fig. 3e (note that the apparent irregularity arises from the fact

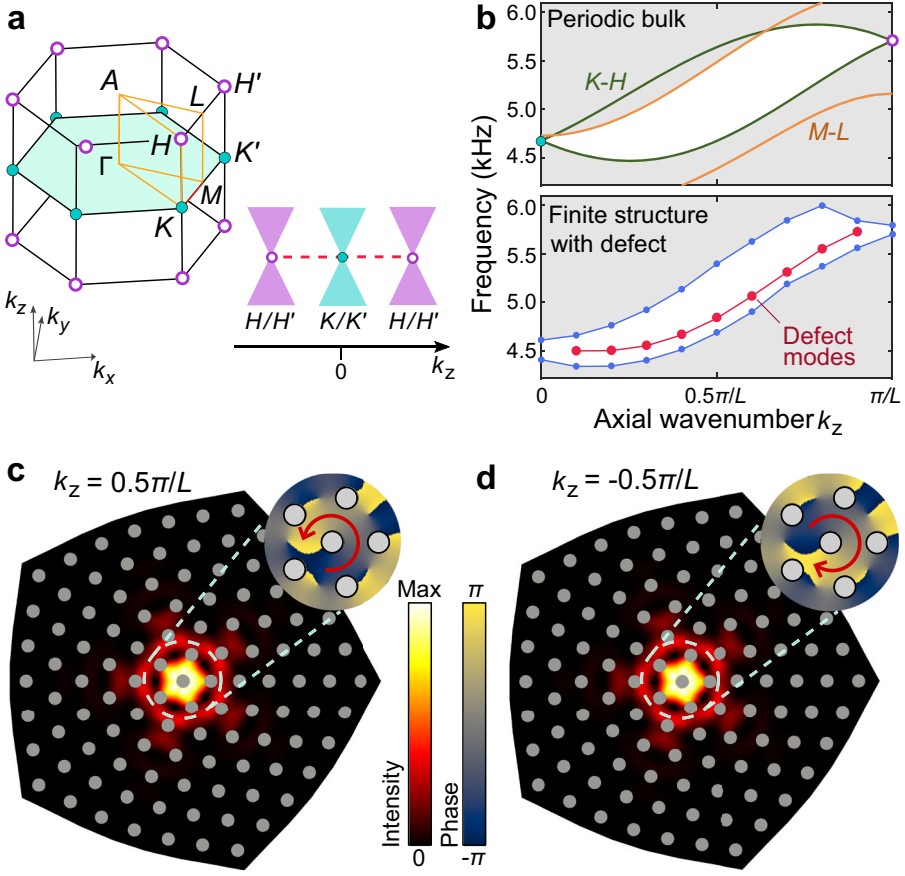

**Fig. 2 Numerical characterisation of the TLD-bound modes. a** Left panel: 3D Brillouin zone of the acoustic crystal without a TLD. Weyl points occur at $K$ and $K'$ with topological charge +1 (cyan dots), and at $H$ and $H'$ with charge −1 (magenta dots). Right panel: projection of the Weyl points onto $k_z$, with red dashes indicating the TLD-bound modes. **b** Numerical bandstructures. Upper plot: bands along $K$-$H$ and $M$-$L$ for the periodic bulk. Lower plot: bands for a structure with a TLD (with the same cross sectional profile as in Fig. 1b–e) and periodicity $L$ along $z$; band edge (in-gap) modes are plotted in blue (red). In-gap regions are shown in white. All bands, including the TLD-bound modes, are symmetric around $k_z = 0$; only the $k_z > 0$ range is plotted here. **c, d** Calculated acoustic pressure intensity distributions in the $x$–$y$ plane, with $z$ at the midpoint of the structure's central air sheet, for the TLD-bound modes at $k_z = \pi/2L$ (**c**) and $k_z = -\pi/2L$ (**d**). Both modes have frequency 4.844 kHz. The grey circles are the solid rods passing through the air sheet. Insets: phase distribution of acoustic pressure near the TLD, showing that the two modes have opposite OAM.

that the measurement points lie at different azimuthal angles). The measurement data is in good agreement with the numerically obtained TLD-bound mode profiles. From a linear least squares fit of the semi-logarithmic plot, using measurement data up to a radial distance of 12 cm, we find a localisation length of 2.38 cm, which is on the order of the mean distance between unit cells (i.e., the approximate lattice constant).

Figure 3f plots the phase of the measured acoustic pressure signal versus azimuthal angle for $k_z = \pi/2L$ and $f = 4.924$ kHz. The different data series in this plot correspond to measurement points at different radial distances. The phase is observed to wind by $+2\pi$ during a counterclockwise (CCW) loop encircling the TLD, consistent with the numerically obtained eigenmode (Fig. 2c), which implies that the TLD-bound mode has OAM of +1.

**Excitation by vortex sources**. To demonstrate the physical significance of the OAM carried by the TLD-bound modes, we studied their coupling to external acoustic vortices. The experimental setup is shown in Fig. 4a. The vortex wave is generated in a cylindrical waveguide of radius 1.7 cm, attached to the bottom layer of the sample at the center of the TLD. Figure 4b shows the acoustic pressure intensity measured in the top layer, on the opposite side of the sample from the source. This intensity is obtained by averaging over points closest to the TLD, and

dividing by the averaged intensity in the bottom layer to normalise away the frequency dependence of the source. For a CCW vortex source, a strong peak is observed within the range of frequencies where TLD-bound modes are predicted to exist. For a clockwise (CW) vortex source, the intensity is low (the non-vanishing intensity is likely due to finite-size effects).

Figure 4c–d shows the intensity and phase distributions measured in the top layer at 5.6 kHz, confirming that the TLD-bound modes are preferentially excited by the CCW vortex.

After the TLD-bound modes have passed through the structure, they emit an acoustic vortex into free space at the far surface. In Fig. 4e–h, we show the intensity and phase distributions measured by an external acoustic probe positioned 2 mm above the top surface of the sample. For a CCW vortex source in the bottom layer, a CCW vortex is emitted from the top layer, at the position of the TLD; for a CW vortex source, the emission is negligible due to the TLD-bound modes not being excited. For frequencies outside the range of the TLD-bound modes, the CW and CCW vortices both produce negligible emission from the top layer (see Supplementary Note 4).

## Discussion

We have experimentally realised a 3D acoustic structure hosting localised topological modes induced by a topological lattice

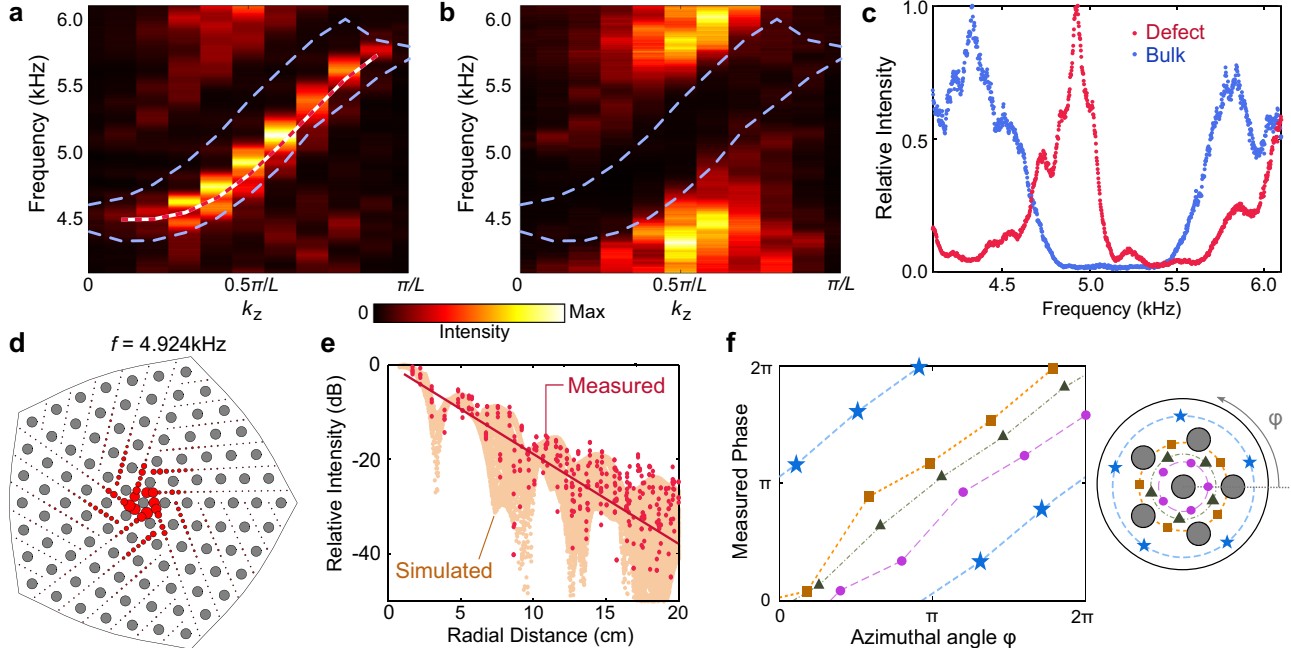

**Fig. 3 Experimental observation of TLD-bound modes. a** Measured spectral intensity of the TLD-bound modes. With an acoustic source in the bottom layer near the TLD, the acoustic pressure readings are Fourier transformed in $z$, and the mean in-plane intensity within 5 cm of the TLD is plotted. Red-and-white dashes show the numerically obtained TLD-bound mode dispersion curve, and blue dashes show the numerically obtained band edges. **b** Spectral intensity of bulk modes, obtained by placing the source and probe 14 cm and 10 cm away from the TLD, respectively. **c** Defect (red) and bulk (blue) spectral intensities at $k_z = \pi/2L$, with each curve normalised to its maximum value. **d** Measured in-plane intensity distribution at $k_z = \pi/2L$ and frequency 4.924 kHz. Each red circle is centered at a measurement point, and its area is proportional to the squared magnitude of the acoustic pressure. **e** Semi-logarithmic plot of intensity versus radial distance from the TLD for the mode in (**d**). Red dots show the experimental results, taken at different azimuthal angles, and orange dots show the profile of the numerically obtained TLD-bound eigenmode. **f** Left panel: measured phase signal versus azimuthal angle $\varphi$ at different radial distances near the TLD. Right panel: schematic of the measurement positions. In **d** and **f**, the grey circles indicate the solid rods.

defect. In real space, the modes lie along a 1D line formed by the defect, embedded within the bulk; in momentum space, they connect the projections of the Weyl points in the defect-free crystal, and hence span the 1D Brillouin zone. This is, to our knowledge, the first experimental demonstration of a defect-induced topological mode in any 3D system. For each momentum space slice ($k_z$), the system maps onto a 2D Chern insulator trapped on a surface with singular curvature. Theoretical studies have previously shown that such a system hosts a robust localised defect mode tied to the Chern number of the 2D bulk bandstructure[8,46,47].

The TLD-bound modes carry nonzero OAM, locked to their propagation direction. This is a striking feature not possessed by topological defect modes based on other similar schemes; for example, the localised topological modes of 2D Kekulé lattices carry zero winding number[67–70]. Our sample therefore serves as an OAM-locked acoustic waveguide, one whose operating principles are very different from the chiral acoustic emitters[71,72] and metasurfaces[73,74] studied in previous works. This design may be useful for applications of acoustic vortices, such as acoustic traps and rotors[48,49] and OAM-encoded communications[50]. Similar designs could be used to realise TLD-bound modes in photonics, based on 3D photonic crystals[30] or laser-written waveguide arrays[35].

Finally, our work opens the door for further investigations into the numerous other effects of lattice defects in topological materials. Many interesting phenomena in this area have been proposed theoretically but have not thus far been observed, including torsional chiral magnetic effects in Weyl semimetals and 1D helical defect modes in 3D weak topological insulators[5,13,15].

## Methods

**Lattice generation**. The lattice was optimised by the the molecular dynamics simulator LAMMPS[75], using two types of particle interactions: (i) a three-body Tersoff potential (`SiC.tersoff`), and (ii) a pairwise nearest-neighbour harmonic potential $U(r) = K(r - r_0)^2$ (`bond_style harmonic`) with $K = 20$ and $r_0 = 0$. Note that these particle interactions have no physical significance; they are simply a convenient way to generate a lattice with minimal variation in inter-site distances[24].

**Numerical simulation**. All bandstructure calculations were performed using COMSOL Multiphysics, with air density 1.18 kg m$^{-3}$ and sound speed 343 ms$^{-1}$. All air-solid interfaces are modeled as hard acoustic boundaries. For the dispersion plot in the lower panel of Fig. 2b, we used periodic boundary conditions in the $z$ direction, and plane wave radiation boundary conditions in $x$ and $y$.

**Experiments**. The experimental samples were fabricated from photosensitive resin via stereolithographic 3D printing. For the dispersion measurements in Fig. 3, the bottom surface of the sample is covered by a square plexiglass plate (length 500 mm), which acts as a hard acoustic boundary. A broadband acoustic signal is launched from a balanced armature speaker of around 1 mm radius, driven by a power amplifier, and located at the center of the TLD at the interface between the plate and the sample. Each acoustic probe is a microphone (Brüel & Kjær Type 4961, of about 3.2 mm radius) in a sealed sleeve with a tube of 1 mm radius and 250 mm length. The probes can be threaded into the sample along the horizontal air regions to scan different positions within each layer of the sample (see Supplementary Note 4). The measured data was processed by a Brüel & Kjær 3160-A-022 module to extract the frequency spectrum, with 2 Hz resolution. Spatial Fourier transforms are applied to the complex acoustic pressure signals to obtain the dispersion relation and field distributions.

For the experiment shown in Fig. 4, the CW and CCW waves are generated in a circular waveguide of radius 1.7 cm, into which three balanced armature speakers are inserted. The signal amplitudes in the three speakers are kept the same, and the phases are controlled by two waveform generators (Agilent type 33500B). The CW and CCW waves were generated by setting the relative phases to (0°, ±120°, ±240°).

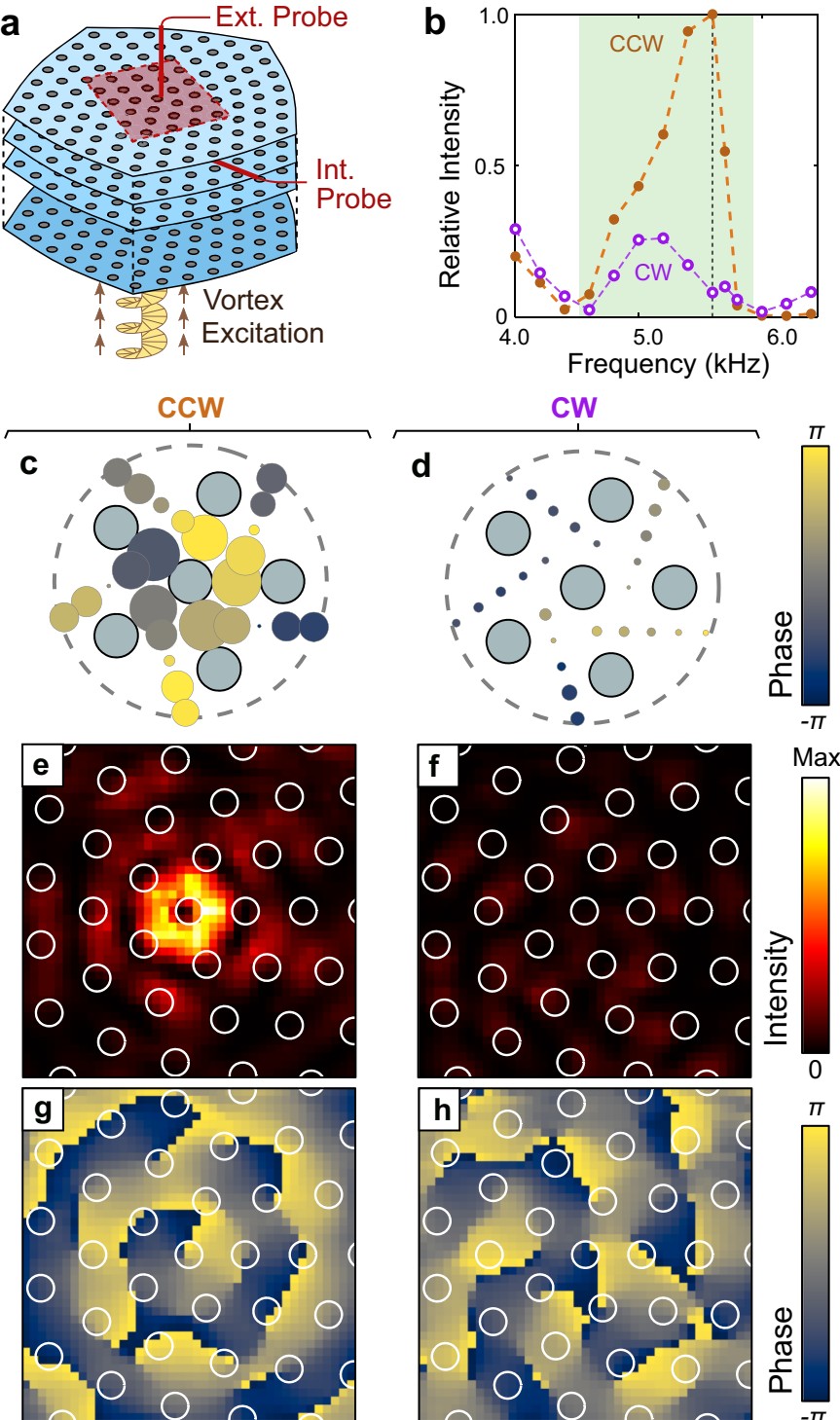

**Fig. 4 Selective excitation by a vortex source. a** Schematic of the experiment. A counterclockwise (CCW) or clockwise (CW) acoustic vortex is incident on the bottom of the sample, centered on the TLD. An external probe, 2 mm above the top surface, sweeps over a 20 cm × 20 cm area (0.5 cm step size). An internal probe is inserted into the top layer. **b** Normalised acoustic pressure intensity versus frequency measured in the top layer by the internal probe for a CCW (purple dots) and CW (orange dots) vortex source. The green region indicates the frequency range hosting TLD-bound modes. **c, d** Acoustic pressure distribution in the top layer for a CCW (**g**) and CW (**h**) vortex source at 5.6 kHz (vertical dotted line in **b**). The area and colour of each circle correspond to intensity and phase respectively. **e, h** Intensity distributions (**e, f**) and phase distributions (**g, h**) measured by the external probe for a CCW (**c**) and CW (**d**) vortex source. The grey circles in (**c–d**) and white circles in (**e–h**) indicate the structural rods.

## Data availability

The data supporting the findings of this study are available from the Digital Repository of Nanyang Technological University (DR-NTU) at https://doi.org/10.21979/N9/THY532.

## Code availability

All numerical codes are available from the corresponding authors on reasonable request.

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

## Acknowledgements
Q.W., H.X., B.Z. and Y.C. acknowledge support from Singapore MOE Academic Research Fund Tier 3 Grant MOE2016-T3-1-006, Tier 1 Grant RG187/18, and Tier 2 Grant MOE2019-T2-2-085. Y.G., H.-X. S., D.J., Y.-J.G. and S.-Q.Y. acknowledge support from the National Natural Science Foundation of China under Grants No. 11774137 and 51779107, National Key R&D Program Project (No. 2020YFC1512403 and 2020YFC1512400) and the State Key Laboratory of Acoustics, Chinese Academy of Science under Grant No. SKLA202016.

## Author contributions
Q.W. and Y.G. contributed equally to this work. Q.W., B.Z. and Y.C. conceived the idea. Q.W. designed the acoustic structures and performed the numerical simulations. Q.W., H.X., and H.-X.S. designed the experiments and fabricated the sample. Y.G., H.-X.S., D.J. and Y.-J.G. conducted the measurements. S.-Q.Y., B.Z. and Y.C. supervised the project. All authors contributed extensively to the interpretation of the results and the writing of the paper.

## Competing interests
The authors declare no competing interests.
