## [Peer Review File · Nature Communications]

REVIEWER COMMENTS

Reviewer #1 (Remarks to the Author):

In this paper by Wang et al. the authors consider topological defects in Weyl systems. In a nutshell it is known that when one has a quantum hall system/topological insulator, fluxes or monodromy defects in general bind modes that directly manifest the underlying topology. Considering a Weyl system and denoting the momentum direction separating the nodes as the perpendicular direction, the planes undergo a phase transition by tuning k_z through the nodes that changes the Chern number. Hence as function of k_z lattice defects then bind modes for those k_z values. This latter is because defects act as torsion [for dislocations] or localized curvature. This gives an effective flux. In fact all these notion only work if the flux is localized [order of the lattice spacing] as one really needs flux defects- TLDs do this naturally but are then harder to grow in materials. In this case however the authors profit from the metamaterial setting to have control over these aspects. And reproduce the effective model using an acoustic design.

The paper is nicely written, apart from some evident omissions in terms of referencing and the data is convincing. Also the theme is rather timely, see January issue of Nature where disclinations in as probes of metamaterials have been featured. There are some small issues to settle before I can give a final verdict;

1] The authors prominently introduce higher order Weyl ideas. In some sense this is related but the defect mechanism really acts on a normal Weyl state just as defects probe normal TIs. The k_z dependence comes from the usual transition, see above and the D-2 correspondence by a defect correspondence in the bulk rather than a hinge per se. So this prominent relation is not as direct as suggested and could also be mentioned after the result.

2]related to 1 the TLD modes are thus not HOFA states indeed the are defect states that do not a hinge geometry that can break symmetries on edge. I think the authors should not mis them with HOFA states. In those nice papers the idea is stack higher TIs in certain fashion such that the polarization mode for each k_z has spectrum that crosses [very much as in a SSH model having mirror] and thus only exist for certain k_z values. Here we are taling about 1st order states and dfect correspondence.

3] With regard to literature the authors have missed some relevant developments. Apart from ref 22 there where two groups that also pointed out these defect correspondences;

A. <https://journals.aps.org/prb/abstract/10.1103/PhysRevB.82.115120>
where the authors use the tenfold way directly

B. <https://www.sciencedirect.com/science/article/abs/pii/S0022369717313665>
which includes a perceptive from Green's functions, see
<https://journals.aps.org/prb/abstract/10.1103/PhysRevB.92.085126>] and also the k_z
decomposition as in the paper , see
<https://journals.aps.org/prb/abstract/10.1103/PhysRevB.90.241403>.

In this regard TLD modes were not only used to manifest the weak indices but also generalized crystalline topologies

C. https://scholar.google.nl/citations?user=LU_C7SMAAAA&hl=nl

Finally the authors might want to know that these modes have been seen in materials;

D. <https://journals.jps.jp/doi/pdf/10.7566/JPSJ.89.023703>

the authors should give fair credit to these earlier works A-D.

4] The analysis on page 3, top right column, is a bit unclear. The effective flux Hamiltonian is fine but then the authors say that TRS partners are on the other segment. This has to be better motivated. Does the system have TRS [not guaranteed for Weyl]? The Hamiltonian is not given. Namely, the two can also be two chiral copies [i.e. the Hamiltonian is block diagonal in that sense.] This has to be made more clear. This will show why they can be treated separately [as should as we really have TRS breaking QHE slices for the k_z values as is also manifest from the effective Hamiltonian].

This analysis has to be updated and made explicit in the SM [not only effective Weyl Hamiltonians] and with a few short statements in the main text. As now the reader has to trust the Hamiltonian is correct and gives effective Weyl systems at the high symmetry points. This is of course the case but the relation between them is important and has to be benchmarked to a total Hamiltonian effectively reproduced, also to check the data.

5] the authors say a few time crystalline weyl-semimetal. The crystalline is implied all the time so does not have to be mentioned.

Reviewer #2 (Remarks to the Author):

In this work, the authors construct an acoustic Weyl metamaterial with disclination defects, and examine the one-dimensional chiral states bound to the defect. They show through a combination of theoretical, numerical, and experimental results that disclination lines in this acoustic semimetal indeed bind topological modes. This is an impressive result of wide interest, representing--to my knowledge--the first experimental confirmation of disclination bound states in a three-dimensional system.

However, before recommending this work for publication in Nature Communications, I think it is important to point out that the one-dimensional defect states synthesized here are conceptually distinct from "higher order" topological modes introduced recently. There is no higher order topological index in this acoustic Weyl semimetal, contrary to works such as Refs. 18 and 19 in the present manuscript. On the other hand, as noted by the authors there is a much deeper connection between the modes presented here and previous work on curvature response in Chern insulators and Weyl semimetals such as:

Phys. Rev. Lett 69, 953
Phys. Rev. D 90, 105004
Phys. Rev. Lett. 110, 046401
Phys. Rev. B 88, 155127
Nature 534, 671--675

It is certainly not the authors fault that "higher order" has come to be a catch-all phrase for subdimensional bound states, but it is important that the present work is distinguished from parallel and conceptually distinct efforts investigating higher order topological materials that do not show surface signatures. Once properly contextualized, I believe this work will be suitable for publication.

Response to Reviewers

We are grateful for the constructive comments from the Reviewers for our manuscript (NCOMMS-20-49188-T, “Vortex higher-order Fermi arc induced by topological lattice defects”). Below, each of the comments from each reviewer is quoted in *italics* and followed by our response. We have also revised the manuscript and the Supplementary Information accordingly, and these updates are highlighted in red in those files.

The revised manuscript contains two particularly notable changes:

1. Based on comments by both Reviewers, we have re-arranged the paragraphs of the introduction to discuss the concepts in a different order. The previous sequence was: (i) high-order Fermi arcs, (ii) topological lattice defects, and (iii) our acoustic meta-material. The revised introduction discusses (i) topological lattice defects, with an expanded section on their relevance to topological materials, (ii) our acoustic meta-material, and (iii) the relationship with higher-order Fermi arcs. We believe this sequence does a better job of explaining the significance of our work, and are grateful to the Reviewers for pointing us in the right direction.
2. Due to concerns by both Reviewers that the modes induced by the topological lattice defect (TLD) should not be called “higher-order” modes, we have changed the terminology. Instead of calling them “higher-order Fermi arc” (HOFA) modes, we now call them “TLD-Induced Fermi Arc” (TIFA) modes.

.....

Reviewer #1

Reviewer Comments: *In this paper by Wang et al. the authors consider topological defects in Weyl systems. In a nutshell it is known that when one has a quantum hall system/topological insulator, fluxes or monodromy defects in general bind modes that directly manifest the underlying topology. Considering a Weyl system and denoting the momentum direction separating the nodes as the perpendicular direction, the planes undergo a phase transition by tuning k_z through the nodes that changes the Chern number. Hence as a function of k_z lattice defects then bind modes for those k_z values. This latter is because defects act as torsion [for dislocations] or localized curvature. This gives an effective flux. In fact all*

these notion only work if the flux is localized [order of the lattice spacing] as one really needs flux defects- TLDs do this naturally but are then harder to grow in materials. In this case however the authors profit from the metamaterial setting to have control over these aspects. And reproduce the effective model using an acoustic design.

The paper is nicely written, apart from some evident omissions in terms of referencing and the data is convincing. Also the theme is rather timely, see January issue of Nature where disclinations in as probes of metamaterials have been featured. There are some small issues to settle before I can give a final verdict;

Response: We thank the reviewer for referring to the work as “*nicely written*”, and for commenting that “*the data is convincing*” and “*rather timely*”. Regarding the other comments about “*some evident omissions in terms of referencing*” and “*some small issues to settle*”, we have made improvements to the paper in response to the Reviewer’s comments, as detailed below.

Reviewer Comments: *The authors prominently introduce higher order Weyl ideas. In some sense this is related but the defect mechanism really acts on a normal Weyl state just as defects probe normal TIs. The k_z dependence comes from the usual transition, see above and the D-2 correspondence by a defect correspondence in the bulk rather than a hinge per se. So this prominent relation is not as direct as suggested and could also be mentioned after the result... related to 1 the TLD modes are thus not HOFA states indeed the are defect states that do not a hinge geometry that can break symmetries on edge. I think the authors should not mis them with HOFA states. In those nice papers the idea is stack higher TIs in certain fashion such that the polarization mode for each k_z has spectrum that crosses [very much as in a SSH model having mirror] and thus only exist for certain k_z values. Here we are talking about 1st order states and dfect correspondence.*

Response: We thank the reviewer for this useful comment, and agree with the reasoning. In the revised manuscript, we no longer refer to the TLD-induced modes “higher-order Fermi arc (HOFA)” modes. Instead, we now call them **TLD-Induced Fermi Arc or TIFA modes**.

We have also re-organized the introduction to avoid wrongly giving the impression that there is a close relationship between the TIFA modes and previously-studied higher-order Weyl Fermi arc modes. The first paragraph of the introduction now starts by introducing topological lattice defects, and the discussion of higher-order Weyl semimetals is now moved

to the third paragraph, with a description of how it differs from the TIFA modes. This paragraph is reproduced below:

“Recently, the discovery of higher-order topological materials⁴¹ has led to the idea of higher-order Weyl and Weyl-like phases^{42–50}, which can host “higher-order Fermi arcs”^{42–47,50}. Like the TIFA modes discussed in this paper, higher-order Fermi arc modes are one-dimensional, but they arise from a completely different mechanism involving higher-order topological indices^{45,46,50}. Moreover, they lie along external hinges, whereas the TIFA modes are localised to the line of the TLD, embedded inside a 3D bulk.”

Moreover, the title of the paper has also been revised to “*Vortex States in an Acoustic Weyl Crystal with a Topological Lattice Defect*”, in order to remove the term “Higher-Order”.

Reviewer Comments: *With regard to literature the authors have missed some relevant developments. Apart from ref 22 there were two groups that also pointed out these defect correspondences;*

A. journals.aps.org/prb/abstract/10.1103/PhysRevB.82.115120 where the authors use the tenfold way directly

B. www.sciencedirect.com/science/article/abs/pii/S0022369717313665, which includes a perspective from Green’s functions,

see journals.aps.org/prb/abstract/10.1103/PhysRevB.92.085126 and also the k_z decomposition as in the paper,

see journals.aps.org/prb/abstract/10.1103/PhysRevB.90.241403.

In this regard TLD modes were not only used to manifest the weak indices but also generalized crystalline topologies

C. scholar.google.nl/citations?user=LU_C7SMAAAAJ&hl=nl

Finally the authors might want to know that these modes have been seen in materials;

D. journals.jps.jp/doi/pdf/10.7566/JPSJ.89.023703

the authors should give fair credit to these earlier works A-D.

Response: We apologize for the omission of these papers.

In the revised manuscript, the first paragraph in the introduction now gives a more detailed background on the literature of topological lattice defects in topological band insulators

and other topological materials. All of the papers referred to by the Reviewer above, and some additional ones, are cited in the revised paragraph. We are grateful to the Reviewer for these pointers.

Reviewer Comments: *The analysis on page 3, top right column, is a bit unclear. The effective flux Hamiltonian is fine but then the authors say that TRS partners are on the other segment. This has to be better motivated. Does the system have TRS [not guaranteed for Weyl]? The Hamiltonian is not given. Namely, the two can also be two chiral copies [i.e. the Hamiltonian is block diagonal in that sense.] This has to be made more clear. This will show why they can be treated separately [as should as we really have TRS breaking QHE slices for the k_z values as is also manifest from the effective Hamiltonian].*

This analysis has to be updated and made explicit in the SM [not only effective Weyl Hamiltonians] and with a few short statements in the main text. As now the reader has to trust the Hamiltonian is correct and gives effective Weyl systems at the high symmetry points. This is of course the case but the relation between them is important and has to be benchmarked to a total Hamiltonian effectively reproduced, also to check the data.

Response: We thank the reviewer for this comment. Our system preserves time reversal symmetry (T). After introducing the topological lattice defect, there is only one good quantum number remaining, i.e. k_z . Given a defect mode at some $k_z > 0$, there must be another defect mode at $-k_z$, which is the counterpart under T .

This is consistent with the analysis based on Eq. (2) in the manuscript. After the relevant gauge transformation is applied, as discussed in the paper, the effective Hamiltonian consists of two sub-blocks. Consider a given k_z , say $k_z > 0$ with $m(k_z) > 0$. Then one of the two Hamiltonian sub-blocks (indexed by $\tau' = 1$) supports a defect mode. For $-k_z$, the other sub-block ($\tau' = -1$) supports a defect mode.

To clarify this point, we have revised the relevant paragraph in **Result Section** (top right column on page 3). The paragraph is reproduced here:

“Hence, viewed from 2D, the pseudo-magnetic flux induces topologically protected chiral defect states. For each $k_z > 0$, one can show^{5,8} that there is a single bound solution (among the two Weyl Hamiltonians) localised at $r = 0$. This remains true even when k_z is non-perturbative. For fixed k_z , the lattice in the

absence of the TLD maps to a 2D Chern insulator whose Chern numbers switch sign with k_z (the gap closes at 0 and $\pm\pi/L$); upon introducing the TLD via the cut-and-glue construction, one of the two sub-blocks in the effective Hamiltonian ($\tau' = 1$ for $0 < k_z < \pi/L$, and $\tau' = -1$ for $-\pi/L < k_z < 0$) exhibits a solution that is localised to the TLD⁸. As we vary k_z , this family of solutions spans the projections of the Weyl points at $K(K')$ and $H(H')$. Note that the overall acoustic structure preserves time-reversal symmetry (T), but the individual Hamiltonian sub-blocks effectively break T ; the defect mode at $-k_z$ thus serves as the time-reversed counterpart of the defect mode at k_z , with opposite chirality. For further details, refer to Supplementary Section II.”

We have also revised the right panel of Fig. 2a (the conceptual schematic of how the TLD-induced modes are related to the Weyl points), to better convey the fact that the TLD mixes the valleys (K and K' , H and H'). The aforementioned two Hamiltonian sub-blocks each consists of a mix of both valleys.

In the Supplementary Information, we have added one sentence in the first paragraph to clarify the role of TRS: *“As the system preserves time reversal symmetry, $m(k_z) = -m(-k_z)$. For convenience, here we suppose that $m(k_z) > 0$ for $k_z > 0$.”* Moreover, we have revised the paragraph under Eq. (S7) as follows: *“For each k_z , there is a single solution arising from one of the two choices of τ' ($\tau' = 1$ for $0 < k_z < \pi/L$, and $\tau' = -1$ for $-\pi/L < k_z < 0$)[2]; as the system preserves time reversal symmetry, the defect mode for $-k_z$ should be the counterpart of the solution at k_z with the same energy but the opposite chirality.”*

Reviewer Comments: *the authors say a few time crystalline weyl-semimetal. The crystalline is implied all the time so does not have to be mentioned.*

Response: We thank the reviewer for raising this issue. In the revised manuscript, we have removed “crystalline” before “Weyl semimetal”.

.....
Reviewer #2

Reviewer Comments: *In this work, the authors construct an acoustic Weyl metamaterial with disclination defects, and examine the one-dimensional chiral states bound to the defect. They show through a combination of theoretical, numerical, and experimental re-*

sults that disclination lines in this acoustic semimetal indeed bind topological modes. This is an impressive result of wide interest, representing—to my knowledge—the first experimental confirmation of disclination bound states in a three-dimensional system.

It is certainly not the authors fault that "higher order" has come to be a catch-all phrase for subdimensional bound states, but it is important that the present work is distinguished from parallel and conceptually distinct efforts investigating higher order topological materials that do not show surface signatures. Once properly contextualized, I believe this work will be suitable for publication.

Response: We thank the Reviewer for referring to the work as "*an impressive result of wide interest*".

Regarding the comment that "*it is important that the present work is distinguished from parallel and conceptually distinct efforts investigating higher order topological materials that do not show surface signatures,*" we have made several improvements according to the Reviewer's comments, as discussed below.

Reviewer Comments: *However, before recommending this work for publication in Nature Communications, I think it is important to point out that the one-dimensional defect states synthesized here are conceptually distinct from "higher order" topological modes introduced recently. There is no higher order topological index in this acoustic Weyl semimetal, contrary to works such as Refs. 18 and 19 in the present manuscript. On the other hand, as noted by the authors there is a much deeper connection between the modes presented here and previous work on curvature response in Chern insulators and Weyl semimetals such as: Phys. Rev. Lett 69, 953; Phys. Rev. D 90, 105004; Phys. Rev. Lett. 110, 046401; Phys. Rev. B 88, 155127; Nature 534, 671–675*

Response: We thank the reviewer for this useful comment. Indeed, the TLD-induced modes are conceptually distinct from the hinge modes discussed in previous papers on higher-order topological semimetals or insulators. To avoid misleading readers, we have made following changes:

1. We have replaced "*higher order Fermic Arc*" (*HOFA*) *modes*" with the terminology "*TLD-induced Fermi Arc*" (*TIFA*) *modes*. We have also changed the title to "*Vortex States in an Acoustic Weyl Crystal with a Topological Lattice Defect,*" in order to

remove “higher-order” from the title.

2. We have rearranged the Introduction section to avoid giving undue weight to higher-order topological semimetals. As discussed at the top of this document, the introduction now starts by discussing topological lattice defects and their implications for topological materials.
3. In the second paragraph, we explain more clearly the reasoning behind the identification of the TLD-induced modes as “Fermi Arc” modes:

“...in a manner analogous to the formation of regular Fermi arcs, the modes span the projections of two Weyl points of opposite topological charge in the axial momentum space k_z . Hence, we refer to them as TLD-induced Fermi arc (TIFA) modes. The TIFA mode for each k_z can be interpreted as a 2D bound state generated by a strongly localised pseudo-magnetic flux associated with the TLD, in accordance with earlier theoretical predictions about disclinations in 2D topological materials.”

4. The third paragraph now explicitly points out that previously-studied higher-order Fermi arc modes *“arise from a completely different mechanism involving higher-order topological indices^{45,46,50}. Moreover, they lie along external hinges, whereas the TIFA modes are localised to the line of the TLD, embedded inside a 3D bulk.”*

We also thank the reviewer for suggesting the additional relevant papers on curvature response, which are now referenced in the revised introduction.

REVIEWER COMMENTS

Reviewer #1 (Remarks to the Author):

The authors have addressed my main concerns of not confusing it with HOFA states. Generally I am satisfied, maybe the authors can address some small issues:

1. I am unsure to call it TIFa modes. These are results of bound states and projected to that space- why do the authors want to connect them in name to arc states?
2. with regard to refs 12-27 <https://www.nature.com/articles/nphys2513> deserves some credit as it generalised weak indices using dislocations results.

the rest of the discussion is up to standard

Reviewer #2 (Remarks to the Author):

In their revised manuscript, the authors have better contextualized their results in terms of the response of topological systems to lattice defects. They have addressed all of my concerns, and I am now confident in recommending this work for publication in Nature Communications.

Response to Reviewers

We are grateful for the constructive comments from the Reviewers for our manuscript (NCOMMS-20-49188-A, “Vortex States in an Acoustic Weyl Crystal with a Topological Lattice Defect”). Below, each of the comments from each reviewer is quoted in *italics* and followed by our response. We have also revised the manuscript and the Supplementary Information accordingly, and these updates are highlighted in red in those files.

.....

Reviewer #1

Reviewer Comments: *The authors have addressed my main concerns of not confusing it with HOFA states.; the rest of the discussion is up to standard.*

Response: We thank the reviewer for the positive comment on the revised manuscript.

Reviewer Comments: *I am unsure to call it TIFa modes. These are results of bound states and projected to that space- why do the authors want to connect them in name to arc states?*

Response: We thank the reviewer for raising this issue. We thought that the “Topological lattice defect (TLD) Induced Fermi Arc” terminology would be helpful for readers focusing on the physics of Weyl media: these modes span the projections of two Weyl points of opposite topological charge, which is intriguingly analogous to the formation of Fermi arcs (both standard or higher-order) in topological semimetals. However, given the reviewer’s understandable reservations over this terminology, we have replaced this with simply “TLD-bound modes” which is more neutral. This change has been made in the revised manuscript and supplementary information.

Reviewer Comments: *with regard to refs 12-27 <https://www.nature.com/articles/nphys2513> deserves some credit as it generalised weak indices using dislocations results.*

Response: We thank the reviewer for suggesting this relevant paper, which is now cited in our revised introduction.

.....

Reviewer #2

Reviewer Comments: *In their revised manuscript, the authors have better contextualized their results in terms of the response of topological systems to lattice defects. They have addressed all of my concerns, and I am now confident in recommending this work for publication in Nature Communications.*

Response: We thank the Reviewer for the positive comment and recommendation of our work.

REVIEWERS' COMMENTS

Reviewer #1 (Remarks to the Author):

I thank the authors for their changes. Indeed the relation to the arc states is subtle see also previous reply for the former improper higher order analogy. In the end, these are defect states, hence my reservations regarding the terminology as in the previous version. The results are very timely and relevant. I am satisfied with changes of this new version and thus now can recommend publication.

Response to Reviewers

We are grateful for the constructive comments from the Reviewers for our manuscript (NCOMMS-20-49188-B, “Vortex States in an Acoustic Weyl Crystal with a Topological Lattice Defect”). Below, each of the comments from each reviewer is quoted in *italics* and followed by our response. We have also revised the manuscript and the Supplementary Information accordingly, and these updates are highlighted in red in those files.

.....

Reviewer #1

Reviewer Comments: *I thank the authors for their changes. Indeed the relation to the arc states is subtle see also previous reply for the former improper higher order analogy. In the end, these are defect states, hence my reservations regarding the terminology as in the previous version. The results are very timely and relevant. I am satisfied with changes of this new version and thus now can recommend publication.*

Response: We thank the reviewer for recommendation of publication.